# Prevalence of Mutations in the *Pfdhfr*, *Pfdhps,* and *Pfmdr1* Genes of Malarial Parasites Isolated from Symptomatic Patients in Dogondoutchi, Niger

**DOI:** 10.3390/tropicalmed7080155

**Published:** 2022-07-29

**Authors:** Ibrahima Issa, Mahaman Moustapha Lamine, Veronique Hubert, Amadou Ilagouma, Eric Adehossi, Aboubacar Mahamadou, Neil F. Lobo, Demba Sarr, Lisa M. Shollenberger, Houze Sandrine, Ronan Jambou, Ibrahim Maman Laminou

**Affiliations:** 1Centre de Recherche Médicale et Sanitaire, Niamey P.O. Box 10887, Niger; ibrarzika@yahoo.fr (I.I.); boubarcas@yahoo.fr (A.M.); rjambou@pasteur.fr (R.J.); 2Faculty of Sciences, University of Zinder, Zinder P.O. Box 656, Niger; laminemahamanmoustapha@gmail.com; 3Centre National de Référence du Paludisme à Paris en France, 75013 Paris, France; hubevero@yahoo.fr (V.H.); sandrine.houze@aphp.fr (H.S.); 4Faculty of Sciences, University Abdou Moumouni of Niamey, Niamey P.O. Box 10662, Niger; ilagoumat@gmail.com (A.I.); eadehossi@yahoo.fr (E.A.); 5Department of Biological Sciences, University of Notre Dame, Notre Dame, IN 46556, USA; neilflobo@gmail.com; 6Department of Infectious Diseases, University of Georgia, Athens, GA 30602, USA; demba.sarr1@gmail.com; 7Department of Biological Sciences, Old Dominion University, Norfolk, VA 23529, USA; lshollen@odu.edu

**Keywords:** *P. falciparum*, *Pfdhfr*, *Pfdhps*, *Pfmdr1*, Niger

## Abstract

The effectiveness of artemisinin-based combination therapies (ACTs) depends not only on that of artemisinin but also on that of partner molecules. This study aims to evaluate the prevalence of mutations in the *Pfdhfr*, *Pfdhps,* and *Pfmdr1* genes from isolates collected during a clinical study. *Plasmodium* genomic DNA samples extracted from symptomatic malaria patients from Dogondoutchi, Niger, were sequenced by the Sanger method to determine mutations in the *Pfdhfr* (codons 51, 59, 108, and 164), *Pfdhps* (codons 436, 437, 540, 581, and 613), and *Pfmdr1* (codons 86, 184, 1034, and 1246) genes. One hundred fifty-five (155) pre-treatment samples were sequenced for the *Pfdhfr, Pfdhps,* and *Pfmdr1* genes. A high prevalence of mutations in the *Pfdhfr* gene was observed at the level of the N51I (84.97%), C59R (92.62%), and S108N (97.39%) codons. The key K540E mutation in the *Pfdhps* gene was not observed. Only one isolate was found to harbor a mutation at codon I431V. The most common mutation on the *Pfmdr1* gene was Y184F in 71.43% of the mutations found, followed by N86Y in 10.20%. The triple-mutant haplotype N51I/C59R/S108N (IRN) was detected in 97% of the samples. Single-mutant (ICS and NCN) and double-mutant (IRS, NRN, and ICN) haplotypes were prevalent at 97% and 95%, respectively. Double-mutant haplotypes of the *Pfdhps* (581 and 613) and *Pfmdr* (86 and 184) were found in 3% and 25.45% of the isolates studied, respectively. The study focused on the molecular analysis of the sequencing of the *Pfdhfr*, *Pfdhps,* and *Pfmdr1* genes. Although a high prevalence of mutations in the *Pfdhfr* gene have been observed, there is a lack of sulfadoxine pyrimethamine resistance. There is a high prevalence of mutation in the *Pfmdr184* codon associated with resistance to amodiaquine. These data will be used by Niger’s National Malaria Control Program to better monitor the resistance of *Plasmodium* to partner molecules in artemisinin-based combination therapies.

## 1. Introduction

The resistance of *Plasmodium falciparum* to antimalarials is a major obstacle to malaria control in most endemic countries. In the face of resistance to chloroquine (CQ), artemisinin-based combination therapies (ACTs) have been recommended by the World Health Organization (WHO) for the treatment of uncomplicated malaria [1]. Currently, ACTs are used in more than 80% of countries that have changed their treatment policy due to the chemoresistance of the parasite [2].

Since the adoption of artesunate–amodiaquine (ASAQ) and artemether–lumefantrine (AL) in Niger, therapeutic efficacy studies (TES) of these two combinations have so far shown satisfactory results [3,4]. However, cases of resistance to artemisinin and its derivatives have already been reported in five Southeast Asian countries: Cambodia, Lao Democratic Republic, Myanmar, Thailand, and Vietnam [5,6]. In addition, a decrease in the sensitivity to ACTs has been observed in several countries, including countries in sub-Saharan Africa [6,7]. Currently, due to the lack of an alternative treatment as effective and efficient as ACTs for the treatment of uncomplicated malaria, enormous efforts must be made to prevent the emergence and spread of resistance to ACTs, especially in sub-Saharan Africa. The effectiveness of artemisinin partner molecules used in ACTs, such as amodiaquine, is a bulwark against the emergence of parasites resistant to artemisinin derivatives. However, it has been clearly demonstrated that mutations in the *Pfmdr1* and *Pfcrt* genes are associated with resistance to aminoquinolines [8,9].

Since ACTs result from a combination of an artemisinin derivative and an associated molecule (lumefantrine for AL and amodiaquine for ASAQ), the effectiveness of the treatment also depends on the sensitivity of the parasite to these partner molecules. As a result, resistance could occur as a result of a reduced sensitivity of the parasite to the latter. For example, since amodiaquine is an aminoquinoline, the hypothesis of an involvement of mutations associated with resistance to aminoquinolines in the mechanism of resistance of *P. falciparum* to ACTs may be possible. In addition, resistance can also be the consequence of selective drug pressure, which has led to the disappearance of susceptible strains (eliminated by treatment) and the emergence of naturally resistant strains.

For almost 10 years in Niger, apart from the management of malaria cases, the National Malaria Control Program (NMCP) has been distributing sulfadoxine pyrimethamine (SP) to pregnant women to prevent malaria during pregnancy [10]. In addition, this molecule (sulfadoxine pyrimethamine) is also used with amodiaquine for the chemoprevention of seasonal malaria in children from 3- to 59-months old in Niger [11]. Clearly, these strategies constitute an additional drug pressure that could promote the emergence of antimalarial-resistant parasites in Niger.

Taken together, these observations support the importance of monitoring molecular markers of resistance of *P. falciparum* to antimalarial compounds used in Niger. In the country, several molecular studies have been conducted to monitor antimalarial resistance, but very few have used samples from symptomatic malaria patients [10]. It is in this context that polymorphisms of the *Pfdhfr*, *Pfdhps*, and *Pfmdr1* genes were analyzed from symptomatic malaria patients.

## 2. Materials and Methods

### 2.1. Sample Source and Study Design

The samples analyzed in this study were collected in 2017 in Doutchi, Niger [4]; only day-0 (pre-treatment) samples were used. Details of the study design were described elsewhere [12]. In short, patients suffering from *P. falciparum* uncomplicated malaria were enrolled, at which time blood samples were taken for microscopic examination and deposited on filter paper (Whatman 3) for subsequent PCR analyses. For the current study, all dried blood spots from day 0 were genotyped to detect SNPs in the *Pfdhfr, Pfdhps*, and *Pfmdr1* genes.

### 2.2. DNA Isolation

Genomic DNA was extracted using the QIAamp DNA Kit (Qiagen, Germany), following the manufacturer’s protocol. The isolated DNA was either used immediately or stored at −20 °C prior to PCR analyses.

### 2.3. SNP Detection of Pfdhfr, Pfdhps, and Pfmdr1 Genes

Point mutations (single-nucleotide polymorphisms, SNPs) of the *Pfdhfr, Pfdhps*, and *Pfmdr1* genes were detected by sequencing. Amplification of the *Pfdhfr, Pfdhps*, and *Pfmdr1* genes was performed using either conventional or nested PCR, as described below.

For *Pfdhfr*, the first PCR (PCR1) used AMP1F (5′-TTTATATTTTCTCCTTTTTA-3′) and AMP2R (5′-CATTTTATTATTCGTTTTCT-3′) primers. The 25 μL PCR reaction mixture contained 2 μL of isolated DNA, 0.25 μL of each primer (5 μM), 3 μL MgCl_2_ (25 mM), 5 μL buffer solution, 2 μL dNTPs (5 mM), 13.25 μL H_2_O, and 0.25 μL Taq polymerase (Invitrogen DNA polymerase). The cycling parameters used an initial denaturation step at 94 °C for 3 min; followed by 45 cycles of denaturation at 92 °C for 30 s, hybridization at 45 °C for 45 s, and extension at 65 °C for 3 min; followed by a final extension at 65 °C for 10 min.

Nested PCR (PCR 2) used AMP2R and AMP3F (5′-TGATGGAACAAGTCTGCGAC-3′) primers. The 25 μL PCR reaction mixture contained 1 μL PCR1 reaction product, 0.25 μL of each primer, 3 μL MgCl_2_, 5 μL buffer solution, 2 μL dNTPs, 13.25 μL H_2_O, and 0.25 μL Taq polymerase. The cycling parameters used an initial denaturation step at 94 °C for 3 min; followed by 30 cycles of denaturation at 92 °C for 30 s, hybridization at 50 °C for 45 s, and extension at 65 °C for 3 min; followed by a final extension at 65 °C for 10 min. The expected length of the amplified *Pfdhfr* gene fragment was 594 bp.

For *Pfdhps*, Dhps-F (5′-TTTTGTTGAACCTAAACGTG-3′) and Dhps-R (5′-AAACGTCATGAACTCTTATTAGAT-3′) primers were used. The 25 μL PCR reaction mixture contained 2 μL of isolated DNA, 0.2 μL of each primer, 4 μL MgCl_2_, 5 μL of buffer solution, 2 μL of dNTPs, 11.35 μL of H_2_O, and 0.25 μL of Taq polymerase. The cycling parameters used an initial denaturation step at 94 °C for 5 min; followed by 40 cycles of denaturation at 94 °C for 30 s, hybridization at 53.1 °C for 30 s, and extension at 72 °C for 1 min; followed by a final extension at 72 °C for 5 min. The expected length of the amplified *Pfdhps* gene fragment was 851 bp.

For *Pfmdr1*, Mdr1-F (5′-TTGCCCACAGAATTGCATCT-3′) and mdr1-R (5′-CGTGTGTTCCATGTGACTGT-3′) primers were used. The 25 μL PCR reaction mixture contained 2 μL of isolated DNA, 0.06 μL of each primer, 3.5 μL MgCl_2_, 2.5 μL of buffer solution, 2 μL of dNTPs, 14.63 μL of H_2_O, and 0.25 μL of Taq polymerase. The cycling parameters used an initial denaturation step at 94 °C for 2 min; followed by 40 cycles of denaturation at 94 °C for 30 s, hybridization at 60.4 °C for 30 s, and extension at 72 °C for 1 min; followed by a final extension at 72 °C for 5 min. The expected length of the amplified *Pfmdr1* gene fragment was 232 bp.

### 2.4. Sequencing of the Pfdhfr, Pfdhps, and Pfmdr1 Genes

Final PCR products were purified using enzymatic cleaning. Briefly, 2 units of exonuclease 1 (USB Corporation, Cleveland, OH, USA), 1 unit of Shrimp Alkaline Phosphatase (USB Corporation), and 1.8 μL of bi-distilled water were added to 8 μL of final PCR amplification products. This mixture was incubated at 37 °C for 15 min, followed by 15 min at 80 °C to inactivate the enzymes.

Each enzymatically purified PCR product was sequenced with forward and reverse PCR primers to generate sense and antisense sequences using Sanger sequencing.

### 2.5. Analysis of Genetic Variability of the Pfdhfr, Pfdhps, and Pfmdr1 Genes

The sequences were aligned and the point mutations (SNPs) were identified using the SeqMan Pro assembler (DNASTAR Inc., Madison, WI, USA). Sequences were manually inspected for quality and sequenced as needed [13,14,15]. The sequences from the clinical isolates were compared to the Pf3D7 wild-type reference strain (GenBank Pf3D7_1343700).

## 3. Results

### 3.1. Characterization of the Pfdhfr, Pfdhps, and Pfmdr1 Genes

One hundred fifty-five (155) samples from day-0 (pre-treatment) patients were sequenced for the genes *Pfdhfr* (codons 51, 59, 108, and 164), *Pfdhps* (codons 436, 437, 540, 581, and 613), and *Pfmdr1* (codons 86, 184, 1034, and 1246). The socio-demographic characteristics of the study population were published previously [4].

### 3.2. Analysis of Simple Mutations in the Pfdhfr, Pfdhps, and Pmdr1 Genes

#### 3.2.1. Prevalence of *Pfdhfr* and *Pfdhps* Alleles

One hundred fifty-five (155) specimens were sequenced for SNPs in the *Pfdhfr* and *Pfdhps* genes. A high prevalence of mutation was observed for the codons N51I (84.97%), C59R (92.62%), and S108N (97.39%) of *Pfdhfr*; however, the mutation at codon I164L was not detected (Table 1). Six SNPs were genotyped in the *Pfdhps* gene at codons I431V, S436A, A437G, K540E, A581G, and A613S. Mutations K540E, S436A, and A437G were not observed, but the other 3 mutations associated with the modulation of parasite responses to sulfadoxine were detected at varying frequencies (Table 1). Only one isolate was found harboring a mutation at codon I431V. The A581G and A613S mutations were detected at 3.03% and 8.82% frequency, respectively.

#### 3.2.2. Prevalence of *Pfmdr1* Gene Mutations

Four point mutations in the *Pfmdr1* sequence (N86Y, Y184F, S1034C, and D1246Y) associated with reduced responses to amodiaquine and lumefantrine were evaluated (Table 1). The most prevalent mutation in the *Pfmdr1* gene (Y184F) accounted for 71.43% of the mutations found, followed by N86Y (10.20%). The S1034C and D1246Y mutations were not detected.

### 3.3. The Prevalence of Multiple Mutations in the Pfdhfr, Pfdhps, and Pfmdr1 Genes

Haplotypes were reconstructed by including all unambiguous data collected at the polymorphic positions designated in the *Pfdhfr*, *Pfdhps*, and *Pfmdr1* genes. *Pfdhfr* double mutant haplotypes, N51I/C59R (IR), N51I/S108N (IN), and C59R/S108N (RN), were common at 95.28%, 97%, and 97.17%, respectively. The *Pfdhfr* triple-mutant N51I/C59R/S108N (IRN) haplotype was detected in 97% of the evaluable samples. The *Pfdhps* double-mutant haplotypes (A581G/A613Y, GY) represent 3% of the samples tested. *Pfmdr1* haplotypes were defined by identifying polymorphic residues at N86Y (Y), Y184F (F), S1034C, and D1246Y. The double mutant haplotype N86Y/Y184F (YF) was found in 25.45% of the sequenced isolates. No *Pfmdr1* genotypes hosting more than two individual mutated codons were detected.

## 4. Discussion

Understanding the mechanism of resistance of *P. falciparum* to ACTs is currently a major concern, as cases of artemisinin resistance have been observed in some parts of the world. This descriptive study highlights the polymorphisms of three malarial genes (*Pfdhfr*, *Pfdhps*, and *Pfmdr1*) from symptomatic malaria patients in Dogondoutchi, Niger in 2017.

Sanger’s method was used to sequence and analyze the genetic variability of these genes. It is a method that remains widely used in sequencing despite the advent of new generations of sequencers (NGS) [16]. PCR/Sequencing coupling is an excellent method of genotyping *P. falciparum* strains.

To evaluate and monitor the effectiveness of antimalarials, molecular markers of resistance can be used [16,17]. It is easier to collect malaria-positive samples from symptomatic patients than from the general population. Samples from symptomatic malaria patients provide enough parasitic DNA to evaluate molecular markers of resistance. In Niger, few molecular studies on antimalarial-drug resistance have used isolates from symptomatic malaria patients [10].

In this study, a high prevalence of mutations was observed in the codons N51I (84.97%), C59R (92.62%), and S108N (97.39%) of *Pfdhfr*. The S108N mutation in the *Pfdhfr* gene is likely to play a key role in pyrimethamine resistance associated with mutations N51I, C59R, and I164L [18,19]. The I164L mutation associated with strong resistance of *P. falciparum* to pyrimethamine [20] was not found in this study. In previous studies carried out in Niger, this mutation has never been so highlighted [10]. The *Pfdhfr* triple mutation (IRN) is strongly associated with resistance to pyrimethamine [21] and is widespread in Africa [22]. Here, it was encountered in 97% of isolates. For years, the increase in the frequency of the IRN triple mutation has been observed in Niger [23,24], although the majority of these studies were carried out on samples collected in the general population [10]. These results could be explained by high drug pressure, as sulfadoxine pyrimethamine has been used in Niger in the chemoprevention of malaria in pregnant women and children for several years [11]. Our results match those found in other countries in Africa [24,25,26,27,28,29]. In countries with high drug pressure, these mutations are the main cause of decreased sulfadoxine pyrimethamine efficacy [30,31]. The S108N point mutation and the IRN triple mutation are essential markers in the evaluation of the effectiveness of sulfadoxine pyrimethamine. In fact, the IRN triple mutation is considered the best predictor of in vivo resistance to sulfadoxine pyrimethamine [30,32].

Mutations affecting the *Pfdhps* gene are responsible for the resistance of *P. falciparum* to sulfadoxine [33]. In previous studies, A437G mutation was found associated with resistance to sulfadoxine pyrimethamine while K540E conferred resistance to sulfadoxine pyrimethamine [34,35]. Our study has not found any mutations in the K540E codon. One would be tempted to say that there is a low level of resistance to sulfadoxine in these parasite isolates. Nevertheless, it should be noted that low prevalence of A581G and A613Y codons have been observed. These mutations have been associated with strong resistance to sulfadoxine pyrimethamine in East Africa [36].

In Niger, sulfadoxine pyrimethamine is also used in ACTs (artesunate plus sulfadoxine–pyrimethamine) available for the management of uncomplicated malaria (The national list of essential drugs from the Department of Pharmacy and Traditional Medicine. Edition 2018, Unpublished document) [4,16]. *P. falciparum* resistance to sulfadoxine pyrimethamine may compromise the effectiveness of the AS-SP combination and lead to the emergence of artemisinin-resistant parasites in Niger.

The molecular characterization of the *Pfmdr1* marker facilitates the prediction of parasite responses to amodiaquine but also to lumefantrine. Parasites combining *Pfmdr1* N86Y and parasites with the *Pfmdr1* triple-mutant haplotype N86Y/Y184F/D1246Y (YFY) have been shown to be resistant to 4-aminoquinolines and are associated with clinical failure with amodiaquine treatment [31,37]. Selection of *Pfmdr1* triple-mutant haplotype (YFY) has been reported during AL treatment and in infections with increased mefloquine sensitivity and reduced susceptibility to lumefantrine [31]. Parasites carrying two or more copies of the wild-type *Pfmdr1* allele are also highly resistant to both lumefantrine and mefloquine [37]. In this study, the Y184F mutation was found to be the most frequent, with 71.43% of mutants, followed by the N86Y mutation. The S1034C and D1246Y mutations were not detected. This is consistent with previous in vitro results, which indicated that clinical samples from Niger respond adequately to other 4-aminoquinolines, such as amodiaquine [38]. Only 10.20% of isolates carried the *Pfmdr*1 mutation N86Y, and the YFY triple-mutant allele was absent. Our results are similar to those obtained from a study conducted in Cambodia [39], which found the *Pfmdr1* mutations Y184F and N86Y at 78.37% and 39.59%, respectively. This high frequency of the *Pfmdr1* Y184F mutation was also reported in Burkina Faso, Senegal, and Gambia [40] with 68.75%, 77.5%, and 51.7%, respectively. However, our results are different from those found in Iran 4 years after the introduction of ACTs (46% for N86Y and 2% for Y184F) [41] and those obtained in the Republic of Congo (73% for N86Y) [42]. These differences could be due to the epidemiological profiles of malaria that vary from one locality to another and at the same time to the strains of *Plasmodium* by their genetic profiles.

The *Pfmdr1* double-mutant haplotype N86Y/T184F (YF) was found in 25.45% of our isolates. Our results were very similar to those obtained in previous studies in Cameroon [43,44].

Amodiaquine is used in Niger for the treatment of uncomplicated malaria but also for the seasonal malaria chemoprevention. The high prevalence of mutations observed in this study could be explained by the drug pressure due to the distribution of millions of amodiaquine tablets each year to children aged 3- to 59-months in Niger, as part of the seasonal malaria chemoprevention [11].

## 5. Conclusions

This study highlighted high prevalence in molecular markers of *P. falciparum* resistance to sulfadoxine-pyrimethamine and amodiaquine, which are molecules used in malaria chemoprevention and ACTs in Niger. There is a need to continuously monitor these markers to predict future *Plasmodium* resistance that may hamper malaria control in the country. A study on the in vivo efficacy of the AS-SP combination will provide a better understanding of the impact of the high frequencies of molecular markers of sulfadoxine pyrimethamine resistance observed on the sensitivity of plasmodia to this molecule in Niger. Future studies should also assess *Pfcrt*. This work has generated essential data for both the NMCP and the Ministry of Public Health of Niger to monitor, control, and guide the fight against malaria in Niger.

## Figures and Tables

**Table 1 tropicalmed-07-00155-t001:** Prevalence of mutations in the *Pfdhfr*, *Pfdhps* and *Pfmdr1* genes.

Gene	Mutation	Genotype	n/N	Frequency (%)
*Pfdhfr*	Single mutant	N51I (I)	130/153	84.97
		C59R (R)	138/149	92.62
		S108N (N)	149/153	97.39
		I164L	0/153	0
	Double mutant	N51I/C59R (IR)	121/125	95.28
		N51I/S108N (IN)	130/134	97
		C59R/S108N (RN)	121/125	97
	Triple mutant	N51I/C59R/S108N (IRN)	121/125	97
*Pfdhps*		K540E	0/33	0
		I431V		0
		S436A		0
		A437G		0
		A581G (G)	1/33	3.03
		A613Y (Y)	3/34	8.82
	Double mutant	A581G/A613Y (GY)	1/33	3.03
*Pfmdr1*	Single mutant	N86Y (Y)	15/147	10.20
		Y184F (F)	105/147	71.43
		S1034C	0/134	0
		D1246Y	0/134	0
	Double mutant	N86Y/Y184F (YF)	14/55	25.45

n: number of samples with the amino acid substitution; N: Total number of samples tested.

## Data Availability

The data is accessible on request at the Center for Medical and Sanitary Research in Niamey. BP: 11887.

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
