# Peer review of "Prevalence of Mutations in the Pfdhfr, Pfdhps, and Pfmdr1 Genes of Malarial Parasites Isolated from Symptomatic Patients in Dogondoutchi, Niger"

_tropicalmed, 2022, doi:10.3390/tropicalmed7080155_

Round 1
Reviewer 1 Report
Prevalence of Mutations in The Pfdhfr, Pfdhps and Pfmdr1 2 Genes of Parasites Isolated During a Therapeutic Efficacy 3 Study of Artemisinin-based Combination Therapies (ACTs) in 4 Doutchi, Niger in 2017
The manuscript shows a molecular study of 155 Pf-infected blood samples, and the sequence analysis of three genes involved in drug resistance: Pfdhfr, Pfdhps and Pfmdr1. The content and context of the manuscript need a better a more careful preparation, revise nomenclature, the presentation of results, and the discussion before it is considered for publication
First of all, in all sections details of the source of samples should not include data that are of no use to describe or discuss the results. E.g. lines 21-22“ DNA samples from a randomized clinical trial comparing the therapeutic efficacy of artéméther lumefantrine (AL) with artésunate amodiaquine (ASAQ) in 2018,” Lines 91-92: and randomly assigned to receive either AL or 91 ASAQ and were followed for 28 days with scheduled visits on days 3, 7, 14, 21 and 28. Lines 189-190: “isolates from a randomized two-arm clinical trial comparing the effi-189 cacy of AL to ASAQ after wide use in Niger.“ It is also suggested to modify the title avoiding “During a Therapeutic Efficacy 3 Study of Artemisinin-based Combination Therapies (ACTs” this should focus on the aim of the present study.
Materials and methods
Lines 87-96, paragraph of section 2.1 must be summarized as there is redundancy, about results of the previous study were reported somewhere else if samples were from day 0 or from patients were given a different ACT treatment.
This section must be mentioned the number of samples analyzed, perhaps geographic distribution, and some other important data to understand better the results.
Line 98: section 2.2, “Parasitic DNA extraction was done” however not only the parasitic DNA was extracted with the commercial kit, it is actually whole DNA from infected blood samples, please clarify.
Section 2.3 please add information of the length of each gene DNA fragment amplified
Results
Please revise, if the title from the subsection is in italics, then the gene abbreviation should not be in italics.
Lines 159-160: I suggest mentioning what protein corresponded to those mutations, also suggest moving “ no mutation…. To the end of the sentence as it was not detected. If write N51I, this is an amino acid substitution, as the codon itself is not written. Please revise. “No mutations were found at codon I164L. A high mutation prevalence was observed in the codons N51I (84.97%), C59R (92.62%), and S108N (97.39%).“
Revise the inconsistency in nomenclature, Y and F are not the gene, are amino acids residues of PfMDR1 (protein). Line 170: “mutation on the Pfmdr1 gene was Y184F“
Check the nomenclature. It is widely accepted when it refers to gene letters in small and in italics, and in capital letters when it refers to the protein. These kinds of errors are written in different sentences of the manuscript. Line 175 “polymorphic positions designated in the Pfdhfr, Pfdhps and Pfmdr1 sequences. Haplotypes, N51I _ C59R, N51I _ S108N and C59R“
Table 1 . What is n/N? no data are provided. There are inconsistencies in the frequencies e.g. why N51L or C59R single mutants were in 84.9% or 92.6, of samples, respectively, and double or triple mutants % is higher? Should be the same % or less, please clarify and revise the values all amino acid substitutions shown in this table. Were the wild parasites, with no mutations, and define wild genotype?
To complete the results, I strongly suggest preparing a table comparing the frequency of amino acid substitutions studied in here and results reported from other geographic areas in Africa.
Discussion
Lines 186-191: this information is part of the methods, also revise to avoid redundancies
Lines 2001-2002: I suggest deleting the following, which seem redundant “In this study, a high prevalence of mutations was observed in the codons N51I 201 (84.97%), C59R (92.62%) and S108N (97.39%), no mutations were found in the I164L codon.”
Revise the following …according to the authors? Or according to the evidence? Lines 222-224: “Indeed, according to the authors, the A437G mutations associated with and K540E confer resistance to SP [35,36]. One would be tempted to say that there is a low level of resistance to sulfadoxin in these parasite isolates.”
Please clarify what haplotype is that? Line 257: “The haplotype for major positions 86 and 184 was found in 74.55% of our isolates”
Please revise, there are some sentences that are confusing, have syntax mistakes and are too long, e.g., Lines 261-264: “However, this result was higher than those found (29.80%; 32%) in a study conducted in Equatorial Guinea by Li et al. [47] on the high prevalence of Pfmdr1 N86Y and Y184F mutations in P. falciparum strains isolated at Bioko's Ile and another study conducted by Ljolie and al. [48] on the prevalence of molecular markers of AL resistance in 3 provinces of Angola.”
how the result was higher? it is more proper to write amino acid substitution than mutations as it is the phenotype. What is the meaning of “the prevalence of molecular markers?
Please revise the syntax, as the prevalence of molecular markers cannot be measured, it might mean certain residue changes? Lines 229-230: this high prevalence in molecular markers of SP resistance may compromise the effectiveness of the AS-SP combination and lead to the emergence of 230 artemisinin resistant parasites and its derivatives in Niger.
Please revise, it might not be necessary to write the author´s name “by Vinayak S. et al. in 2010 “ or “by Dieye and al “ in the sentence, it could be seen in the reference list and its addition makes a long sentence, was relevant data are hidden. Also, revise or rephrase the following text. Lines 245-249: “Our results are similar to those of a study conducted in Cambodia by Vinayak S. et al. in 2010 [41], which had found the mutations Y184F, N86Y respectively 78.37% in the West of Cambodia (Y184F) and 39.59%. This high frequency of the Y184F mutation was reported in Burkina Faso in 2016 by Somé and al. and in Senegal and the Gambia by Dieye and al. [42] 68.75%; 77.5% and 51.7%, respectively.“
Please revise the following sentence as the meaning is confusing: the amplification of mutations A and G? …A is a mutation? ..is a limitation of what? Line 271: “The non-amplification of mutations in the S436A and A437G Pfdhps codons is one of the limitations of this study.”
The authors mentioned that samples came from a clinical study already published, however, they do not mention or integrate that information to the results. In Ibrahima et al. 2020 is mentioned that ArL and ASAQ were effective and well tolerated.
Conclusions
Same problem as before, revise meaning and syntax. if a molecular markers perse can be at high prevalence?. Line 276: “This study highlighted high prevalence in molecular markers of SP and Amodiaquine, which….”
Other Minor
Revise the manuscript thoroughly as the gene abbreviations, parasite genus-species need to be italicized; when referring to the gene
e.g.
line 25, 153, 232, etc : change “Pfmdr1” to italics
line 36,152,157, etc : change “Pfdhfr” to italics
line 68, 186, 205, 222, etc : change “P. falciparum” to italics
line s121, 152,157, etc: change “Pfdhps” to italics
Line 204: revise referencing in the man text “[19] [20].” Or line 147 “[13][14][15].“
Line 219: change “in vivo“ to italics
Line 158: change “thes“ to “these“ or “the”
Revise misspelling: line 128: change “Pfmd1” to “Pfmdr1“(italics)
Line 116: need a comma or revision? “3 μL MgCl2 5 μL buffer solution“
Line 128, 129: please de consistent, and suggested to delete the middle dash from the nucleotide sequence, and revising spaces for all of the sequences of the primers.
Also, check commas and spaces throughout the text.
Need intensive syntax and English revision
Author Response
Attached is the point-by-point response to the reviewers' questions. You will also find the manuscript with the changes made. Thank you very much for your excellent work.

Reviewer 2 Report
This study describes genotyping of pre-treatment Plasmodium falciparum samples from a therapeutic efficacy study performed in Niger. The authors assess three genes (pfmdr1, pfdhfr and pfdhps) for mutations associated with resistance to antimalarial agents. While the characterisation of the resistance mutations in these genes for Niger is of interest, the association of two of the genes (dhfr and dhps) with artemisinin-based combination therapies is unclear. See comments below.
Lines 20-21: The manuscript states that the study aims to evaluate molecular resistance to artemisinin partner molecules, but the pfdhfr and pfdhps genes are not implicated in resistance to either amodiaquine or lumefantrine – the partner drugs assessed on the therapeutic efficacy study. Also, wild type or mutant pfcrt has been implicated in the susceptibility to both amodiaquine and lumefantrine. Why has this gene not been analyzed in this study if that was the aim? This statement about the study aims will need to be re-assessed.
Line 98: Do you mean QIAamp instead of Qiamp? And could you provide the catalogue number of the particular DNA extraction kit you used?
Line 106-134: you provide only the volumes of the reagents used in the PCR reactions. It would be useful to know the final concentrations of the reagents (e.g., MgCl2, dNTPs, primers) and units of Taq polymerase enzyme.
Line 151: What are J0 patients? Does this mean pre-treatment? If so, can you simply state this rather than bring in a new term?
Line 154/155: Could you please delete the words ‘3.1. Subsection’?
Line 158: please correct ‘thes’, perhaps to ‘were sequenced for the pfdhfr and pfdhps genes.’
Line 165: ‘Mutations in the S436A and A437G codons did not amplify…’. You mention in lines 162-163 that all other mutations associated with modulation of parasite responses to sulphadoxine were detected, so this is in conflict with the statement in line 165.
Line 168: Four point mutations were detected but only two are mentioned: Y184F and N86Y. No others are reported. Please explain.
Lines 173-183 (Section 3.3): I find this section very difficult to interpret with respect to the frequencies being listed. What is the denominator for these percentages. For instance, for DHFR, N51I is present in 84.97% of observed codons but is present in the triple mutant at 97%. How can this be. What sample numbers are being used to get these data?
Line 186: Do you mean ACTs, not CTAs?
Line 188: Again, I am unclear as to how polymorphisms in pfdhfr and pfdhps contribute to our understanding of artemisinin or partner drug resistance. These genes are not implicated in resistance to artemisinin derivatives, nor partner drugs used in this TES study.
Lines 214-215:’Alleles of the Pfdhfr gene carrying mutations may cause a delay in parasite clearance.’ This sentence should be deleted. To my knowledge, Pfdhfr mutations have not been implicated in artemisinin resistance related to delayed parasite clearance. So this sentence is confusing.
Lines 228/229: This sentence could be important and justification for assessing the dhfr and dhps genes for resistance in this study in Niger. However, the references do not refer to SP being used in ACTs in Niger. Ref 4 reports on the safety and efficacy of AL and ASAQ, and ref 17 refers to WHO guidelines. How much is AS-SP actually used across Niger, compared to AL? Also, I am not familiar with the abbreviation AS-MS? What is this?
Lines 238/239: As you do not measure gene copy number in this study, perhaps you could omit this sentence.
Line 257: “The NF haplotype……”
Line 263: Replace Bioko’s Ile with Bioko island.
Line 277-279: Also mention the need to assess pfcrt in future surveillance. This is important for both amodiaquine and lumefantrine susceptibility but is completely ignored in your study.
Lines 279-282: Again, this seems reasonable if AS-SP is widely used but I cannot find data that this is the case in Niger. If you have this, then please reference it. Given that SP is used widely in various chemoprevention strategies, I would hope AS-SP is not often used and the current trials comparing efficacy of AL and ASAQ are certainly appropriate.
In Table 1 (row 2), you list Mutants, n/N and Frequencies (%). I can see the mutants and frequencies reported but not the n/N. This will be useful to include.
References: some e.g., 40, 41 and 43 need correction.
Author Response

(The authors gave the same response as above.)

Round 2
Reviewer 1 Report
Prevalence of Mutations in the Pfdhfr, Pfdhps and Pfmdr1 Genes of Parasites Isolated from Doutchi, Niger in 2017 by Issa et al.,
Authors responded to most questions, however, the manuscript needs careful revision, especially some parts e.g. abstract, results/discussion, and other issues in the text. Some concerns were arisen in the previous revision and were not revised.
the abstract:
(line 19) Change “collected in clinical study” to “collected in a clinical study”
(line 20) “Genomic DNA samples extracted from symptomatic malaria patients” It is inaccurate, since genomic DNA was extracted from P.falciparum infected blood samples from symptomatic patients. I suggest to rephrase the sentence.
(lines 20-23) “ were sequenced by the Sanger method to determine mutations in the Pfdhfr genes (codons 51, 59, 108 and 21 164), Pfdhps (codons 436, 437, 540, 581 and 613), and Pfmdr1 (codons 86, 184, 1034 and 1246). One hundred and fifty-five (155) pre-treatment samples were sequenced for the Pfdhfr, Pfdhps and 23 Pfmdr1 genes. A” ….In the sentence the “sequencing” is duplicated, first indicating the codons of interest and secondly the genes. I suggest to rephrase the sentence to be concise and clear.
(lines 29) “Haplotypes I CS and NC N (single mutants) and IR S, N RN and I C N (double mutants) 29 were prevalent at 97% and 95%, respectively.” It seems to be for pfmdr1, and describes prevalence of haplotypes with double mutants? Because there is another sentence on line 31 “ Pfmdr1 haplotypes carrying a double mutation were found in 74.55% of 31 the isolates studied.” Please clarify, as seems that different values are for pfmdr1 double mutants.
(Lines 32-33) found this sentence redundant, please revise:“ The study focused on the molecular analysis of the sequencing of the Pfdhfr, 32 Pfdhps and Pfmdr1 genes. A high prevalence of mutations in the Pfdhfr gene has been observed.”
Results/discussion:
In my opinion one main problem is to sort out point mutations vs amino acid substitutions, as in section 3.2, 3.3, Table 1 and discussion. In table 1 using the title “Table 1. Prevalence of Mutations in the Pfdhfr, Pfdhps and Pfmdr1 genes.“ I would expect to find nucleotide changes in the codons (genes). However, there are described the amino acid variations, and to be consistent, I suggest to change to “Table 1. Prevalence of amino acid substitutions in the PfDHFR, PfDHPS and PfMDR1.” The same problem is observed in the text and is not a minor issue. With the aim to produce a high-quality manuscript, I encourage the authors to make a detailed and careful review of terms as: gene, peptide, mutation, codon, amino acid substitution, nucleotide, and how to use each one.
Line 158, I suggest to delete “DHPS” from “ of the K540E DHPS codon”
Lines 147-9: I suggest to delete the following as is redundant, it is mentioned as if only the codons xx and xx and so on were sequenced. However, it was a gene fragment that comprised codons mentioned. “One hundred and fifty-five (155) samples from day 0 (pre-treatment) patients were sequenced for the genes Pfdhfr (codons 51, 59, 108 and 164), Pfdhps (codons 436, 437, 540, 148 581 and 613), and Pfmdr1 (codons 86, 184, 1034 and 1246). The socio-demographic characteristics of the study population have already been published (4).“
line 152 might be inaccurate “ 3.2.1. Prevalence of Pfdhfr and Pfdhps alleles” as simple mutations are described (check the section: “3.2. Analysis of simple mutations in the Pfdhfr, Pfdhps and Pmdr1 genes“)
There are other problems e.g. line 212 “The point mutation of the 108 DHFR codon “ I suggest to indicate the mutation (e.g. codonX108codonY, or 108codonY) and write it properly. Check other work on this field. It cannot be written only the number of the codon, amino acid or residue position, it should be accompanied by the letter code.
From previous revision:
Section 2.3 please add information of the length of each gene DNA fragment amplified. Revise incomplete sentence on line 123 “the length of Pfdhps DNA fragment amplified was ……”
Check spaces, misspelling, reference´s citation, etc ; e.g. line 84, 104 and others: “Niger[4].”, “MgCl2(25mM)“; Line 130: change “5 Minutes” to “5 minutes”; Revise (line 143): “[13][14][15].”
Author Response
Hello editor in chief of MDPI
Find attached the second round of corrections to manuscript number: tropicalmed-1766831 and entitled: 'Prevalence of Mutations in The Pfdhfr, Pfdhps and Pfmdr1 Genes of Parasites Isolated During a Therapeutic Efficacy Study of Artemisinin-based Combination Therapies (ACTs) in Doutchi, Nigeria in 2017'
The language was corrected by a native English speaker. All the corrections requested by the reviewers have been added and tracked to identify them. Thank you for your excellent editing. Cordially
